# Association between Muscle Fatigability, Self-Perceived Fatigue and C-Reactive Protein at Admission in Hospitalized Geriatric Patients

**DOI:** 10.3390/ijerph20166582

**Published:** 2023-08-16

**Authors:** Carmen Hoekstra, Myrthe Swart, Ivan Bautmans, René Melis, Geeske Peeters

**Affiliations:** 1Department of Geriatric Medicine, Radboud University Medical Centre, Geert Grooteplein Zuid 10 (Route 696), Postbus 9101, 6500 HB Nijmegen, The Netherlands; carmen.hoekstra@ru.nl (C.H.);; 2Gerontology Department, Faculty of Medicine and Pharmacy, Vrije Universiteit Brussel, 1090 Brussels, Belgium; 3Frailty in Ageing Research (FRIA) Group, Vrije Universiteit Brussel, 1090 Brussels, Belgium; 4Department of Geriatrics, Universitair Ziekenhuis Brussel, 1090 Brussels, Belgium

**Keywords:** hand strength, old age, intrinsic capacity, frailty, muscle fatigue

## Abstract

**Background:** The capacity to perceived vitality (CPV) ratio is a novel measure for intrinsic capacity or resilience based on grip work and self-perceived fatigue. CPV has been associated with pre-frailty in older adults and post-surgery inflammation in adults. To better understand the utility of this measure in a frail population, we examined the association between CPV and inflammation in geriatric inpatients. **Methods:** Data were obtained from 104 hospitalized geriatric patients. The average age of participants was 83.3 ± 7.5 years, and 55.8% of participants were women. In the cross-sectional analyses, associations between C-reactive protein (CRP), grip work (GW), self-perceived fatigue (SPF) and the CPV ratio (higher values indicate better capacity) were examined using linear regression adjusted for confounders. **Results:** The adjusted association between CRP (abnormal vs. normal) and the CPV ratio was not statistically significant (B = −0.33, 95% CI = −4.00 to 3.34). Associations between CRP and GW (B = 25.53, 95% CI = −478.23 to 529.30) and SPF (B = 0.57, 95% CI = −0.64 to 1.77) were also not statistically significant. Similar results were found in unadjusted models and analyses of cases with complete data. **Conclusions:** In this frail group of geriatric inpatients, inflammation, routinely assessed with CRP, was not associated with CPV or its components, GW and SPF. Further research is needed to explore whether CPV is a useful indicator of frailty or recovery capacity in hospitalized geriatric patients.

## 1. Introduction

The current trend in gerontological research is to move from a focus on negative concepts of frailty to more positive concepts of resilience and intrinsic capacity [1]. Frailty in older adults is a condition that is defined by a decrease in the spare capacity of the body’s physiological systems [2,3]. Resilience is defined as mental and/or physical recovery after a health stressor [4]. The ability to predict resilience could potentially guide care for older, fragile patients [4,5]. Intrinsic capacity is defined as the combined physical and mental capacities of an individual [6,7]. To assist clinicians in implementing these concepts to facilitate personalised care, tools are needed to quantify the concepts of resilience and intrinsic capacity [8,9]. In recent years, several tools have been developed, which are typical composite scores of the five domains of intrinsic capacity, including locomotion, cognition, sensory, psychology and vitality [10,11]. For the first four domains, a variety of tests are available, but how best to measure vitality remains unclear.

An alternate novel candidate measure is the capacity to perceived vitality (CPV) ratio [2,12]. CPV is calculated as grip work divided by self-perceived fatigue (SPF) [2,12]. Grip work is a measure of muscle fatigability, and measured as grip strength sustained over time. Thus, CPV combines measures of grip strength and fatigue, which are indicators of the frailty phenotype [13]. As such, CPV may be a useful measure of changes in health status or recovery capacity. The CPV ratio suggests that a combination of low grip work and high SPF is indicative of low recovery capacity. Indeed, low CPV values have been associated with pre-frailty in community-dwelling older adults [2]. However, it remains unclear if the new CPV measurement is also a good measure of intrinsic capacity in geriatric inpatients, a vulnerable group with multimorbidity and poor health status. In this group, care tends to be focused more on maintaining functional independence than cure [14]. To facilitate personalised function-focused care, it is particularly important to be able to have function-based measures to predict recovery capacity and inform clinical decision making.

Particularly in frail older adults, inflammation may be a reason for admission or may arise during admission [15]. It may also be a reason for prolonged hospital stay. Inflammation has also been associated with frailty, fatigue and muscle weakness [9] in hospitalised patients [16]. In patients (aged 18+ years) presenting four days post-surgery, both low muscle fatigability and high self-perceived fatigue were associated with the highest surgery-induced inflammation [12]. It is therefore expected that inflammation is also associated with CPV in frail older adults. C-reactive protein (CRP) is primarily classified as an acute indicator of inflammation. In cases of injury, infection or inflammation, levels of CRP can rise significantly [17].

To better understand what CPV measures in geriatric inpatients, we aimed to investigate the cross-sectional associations between inflammation (measured as C-reactive protein, CRP) and CPV, grip work and SPF in hospitalized geriatric patients. We hypothesised that higher CRP levels are associated with lower grip work, higher SPF, and thus lower CPV values.

## 2. Materials and Methods

### 2.1. Study Sample

Data were used from the Geriatric Resilience Registry (GRR), an ongoing registry of patients admitted to the geriatrics ward at the Radboud University Medical Centre, Nijmegen, The Netherlands (see Figure 1). The registry commenced in 2020 with the aim of developing measures to quantify resilience. Within 48 (week days) to 72 (weekend days) hours of admission, patients were invited to participate in the registry. Inclusion criteria were being 65 years and older, an expected length of stay of 3 or more days, permission of the responsible physician to participate, and sufficient understanding of the Dutch language. Patients who were not instructible, had a life expectancy of less than two weeks, or who were in (COVID-19) isolation were excluded from participation. Information regarding instructability and life expectancy of less than 2 weeks were based on the judgement of the care staff and copied from medical records. Informed consent was signed by all participants and their legal representatives in case of reduced capacity. The study was reviewed by the research ethics committee of the Radboud University Medical Center. It did not fall within the remit of the Medical Research Involving Human Subjects Act (WMO). The ethics committee approved the study based on the Dutch Code of Conduct for Health Research, the Dutch Code of Conduct for Responsible Use, the Dutch Personal Data Protection Act and the Medical Treatment Agreement Act (approval number 2021-13022).

The current cross-sectional analyses included data for 104 participants with valid data regarding grip work, self-perceived fatigue and CRP. As a rule of thumb for observational association studies, a minimum of ten participants are required per variable in the model. With a sample of 104 participants, we could add a maximum of 10 variables in the models.

### 2.2. Data Collection

At baseline, The Older Persons and Informal Caregivers Survey (TOPICS) questionnaire and the brief resilience scale (BRS) were completed within 48–72 h of admission. TOPICS is a minimal data set including questions about demographics, health, daily functioning and wellbeing [18]. The BRS measures recovery from stress (range 1–5), with higher scores indicating better self-perceived psychological stress resilience. Ref. [19] Grip work and SPF were measured at baseline and subsequently daily until discharge. Additional information was obtained from electronic patient records, including weight, clinical frailty scale scores (range 1–9) [20], multimorbidity (range 0–17) and length of stay. Reasons for admission were derived from medical records and categorized as “multiple admission reasons”, “general malaise”, “musculoskeletal problems with a fracture”, “musculoskeletal problems without a fracture”, “cognitive problems”, “cardio-vascular problems”, “infections”, “oncological reasons”, and the category “others”. CRP measurements were extracted from laboratory reports. CRP and CPV were not always measured on the same day, which resulted in variation in the time between measurements. The time difference in days between CRP and CPV measurements was calculated.

### 2.3. CPV Ratio

Using previously described methods, the capacity to perceived vitality (CPV) ratio was calculated as [2]:

CPV = GW_weight_/SPF, with higher scores indicating better capacity.

Grip work (GW) is a measure of muscle fatigability and was calculated as [16]:

GW_weight_ = (maximum grip strength × 0.75 × fatigue resistance)/weight.

GW was measured using an Eforto device^®^ [21], which consists of a rubber bulb connected to a smartphone app via Bluetooth. To measure maximum grip strength (kPa), participants maximally squeezed the bulb for five seconds. The maximum values of three attempts were used as the maximum grip strength. Next, to measure fatigue resistance, participants were instructed to maximally squeeze the bulb and maintain this for as long as possible, until instructed to stop. The time (in seconds) that each participant maintained the grip strength until it dropped below 50% of the maximum grip strength was measured. Participants were instructed to use their dominant hand if possible, with thirty seconds rest between measurements. The GW was divided by body weight (kg) to account for weight-related differences in grip strength [21].

Although in other studies of this topic other tools, such as the MFI-20, were used to measure fatigue, we used the single question: “How tired do you feel at this moment?”, to measure self-perceived fatigue (SPF). This question was asked prior to the grip work assessment. Response options ranged from 0 (not tired at all) to 10 (extremely tired). The scores were recoded as 1–11 to avoid division by zero in the calculation of CPV [2].

GW and SPF were measured twice daily while admitted to hospital. In the current analyses, only the first measurement was used (baseline), which was often conducted in the morning.

### 2.4. CRP

CRP measurements were extracted from laboratory reports; blood samples were taken as part of routine clinical care upon admission. CRP levels in the blood were often measured on the first day of admission. In the current analyses, CRP was used as a dichotomous variable. The presence of inflammation or infection was defined as a CRP level exceeding 10 mg/L, based on the hospital laboratory’s reference values.

### 2.5. Statistical Analysis

Descriptive statistics were used to describe the study sample. Cross-sectional analyses of associations between CRP and GW, SPF and CPV were examined using linear regression models using the statistical software RStudio version 3.6.1. Assumptions for linear regression were tested, and these assumptions were met after dichotomization of CRP. The model was run with and without adjustment for confounding variables. Based on the literature and clinical reasoning, the following variables were considered as confounders: age, sex, the BRS, the clinical frailty scale, multimorbidity, length of stay and the time (in days) between CPV and CRP measurements [2,22]. Variables were considered confounders and included in the model if they were associated with both CRP and CPV and led to a >10% change in the regression coefficient after adding the confounder to the model in a forward selection process. The level of significance was set at 0.05, and 95% confidence intervals (CI) were reported.

As there are sex differences in grip strength and the perception of fatigue, the association between CRP and CPV could be modified by sex. To test this, we first examined whether the interaction term (CRP × sex) was statistically significantly associated with CPV. Second, the association between CRP and CPV was fitted for stratification by sex. If the interaction term was statistically significant and the stratified analyses suggested different associations for men and women, further analyses were performed for men and women separately.

Eighteen (17%) participants had missing BRS values; there were no missing values for other variables. Comparison of characteristics between participants with complete and missing data showed that data were not missing completely at random (Appendix A, Appendix A). Hence, missing values were imputed using multiple imputation by changed equations (MICE) [23]. Ten additional datasets were created. Results were pooled using Rubin’s Rules. Both results from complete cases data and pooled results after imputation are presented.

## 3. Results

Data from 104 participants (mean age 83 ± 8 years; 56% women) were included in the current analyses (Table 1). The sample was fairly frail, with a mean clinical frailty scale of 5.5 ± 1.4 and a median number of 6 (IQR 5–6) chronic conditions. The most common reasons for admissions were infection (*n* = 28), musculoskeletal problems with fracture (*n* = 18) and other reasons (*n* = 19). The median CRP level was 26.5 (IQR 4.8–60.0) mg/L, and 63 (61%) participants were classified as having abnormal CRP values. The median CPV value was 1.7 (IQR = 0.9–4.2). Participants with missing values for confounding variables (*n* = 18) did not differ from participants without missing values (*n* = 86) for any baseline characteristic except CRP (*p* = 0.04) (see Appendix A).

### 3.1. Cross-Sectional Association between CRP and CPV

No evidence was found for effect modification by sex (*p*-value interaction term = 0.62; women: B = −1.18, CI = −2.70 to 0.33; men: B = −0.48, CI = −8.31 to 7.35). Participants with abnormal CRP values had 1.69 (CI = −4.48 to 1.10) points lower CPVs (Table 2). However, this association was not statistically significant. After adjustment for confounders, and particularly after imputation of missing values, the association attenuated and remained not statistically significant.

### 3.2. Cross-Sectional Association between CRP and GW and SPF

Participants with abnormal CRP values had 188 (Cl = −680 to 305) lower GW scores and 0.5 (Cl = −0.7 to 1.6) higher SPF scores (Table 2). However, these associations were not statistically significant. After adjustment for confounders and imputation of missing values, the associations remained not statistically significant.

## 4. Discussion

This study examined the association between CRP and CPV in hospitalized geriatric patients. In contrast with our hypotheses, no statistically significant association was found. Moreover, no statistically significant associations were found between CRP and CPV components GW and SPF. A likely explanation for the lack of associations is the homogeneity of the sample.

Our main finding that CRP was not associated with CPV or GW was in line with a previous study that examined the association between CRP and GW in relatively vital community-dwelling older adults [2]. In contrast, a study among older adults without inflammation found a moderately strong correlation between IL-6 and fatigue resistance (r = 0.44, *p* < 0.05), but in men only [24]. It may be that this association between inflammation and CPV is stronger in men than in women. Indeed, a systematic review of 168 studies found a slightly higher correlation between CRP and grip strength in men (r = −0.12) than in women (r = −0.08) [25]. No evidence was found for effect modification by sex in the current analyses, but our study may have been underpowered to detect this. Alternatively, it may be that the association is stronger in people without acute inflammation than in people with acute inflammation. Repeating the main analysis excluding the 28 participants who were admitted because of an infection showed a similar non-significant association between CRP and CPV (*n* = 76, b = 0.78, CI = −3.6, 5.5). However, post hoc analyses of our data show that, indeed, the association appeared to be stronger between continuous CRP and CPV in participants without inflammation (CRP < 10 mg/L: *n* = 41, b = −81.6, CI = −189.0, 25.9) than in participants with inflammation (CRP > 10 mg/L): *n* = 63, b = −0.5, CI = −4.2, 3.1). Note that these associations were not statistically significant, and the numbers per subgroup were small. In contrast, the above mentioned systematic review found that the correlation between CRP and grip strength was higher in geriatric patients (r = −0.19) than in community-dwelling older adults (r = −0.09), but the correlations were only low to modest, and not adjusted for confounding variables [25]. Finally, in the current sample, the majority of patients had abnormal CRP values and low CPV values, suggesting that the sample may have been too homogenous to detect an association. Overall, it remains unclear if CRP is associated with CPV in older adults in general, but particularly in hospitalized geriatric patients.

Inflammation has been linked with perceived fatigue in both high grade chronic conditions (e.g., rheumatoid arthritis, chronic bowel disease) [26] and in low grade chronic inflammation in older adults [27,28]. However, whether inflammation is also associated with fatigue in hospitalised older adults is less clear, as elevated levels may reflect multimorbidity or frailty rather than acute inflammation [29]. The current findings do not provide evidence for such an association, but this should be verified using a different sample.

Strengths of this study include relatively unique data with standardized, digitalized grip work measurements, which add to the study’s reliability [21]. This is one of the first studies to measure CPV in acutely ill hospitalized older patients, a group in which it is particularly important to identify measures of recovery capacity to inform clinical decision making. An important limitation is that participants with missing data regarding CPV had to be excluded. Reasons for missing CPV data were an inability to complete the fatigue resistance test due to low muscular endurance, and refusing study participation due to feeling too tired or too ill. This resulted in a relatively selective, homogenous sample of participants who were acutely ill, but with sufficient energy to be willing to participate and complete the test and questionnaires. This may have resulted in limited statistical power to detect associations and limited generalizability of the findings. Verification of the current findings in larger, more heterogenic samples is required. A second limitation is that CRP and CPV were not always measured on the same day. The median time between these measurements was 1 (IQR 1–2) day. Blood samples for CRP assessment were taken upon admission, while the baseline survey was completed within 48 (weekdays) or 72 (weekend days) hours of admission. However, the half-life time for CRP is estimated to be 19 days, irrespective of its circulating concentration [30]. Thus, resolution of inflammation status might have been missed, given the low-responsiveness of CRP. We planned to also investigate change in CRP and CPV over time, to examine its longitudinal association. However, as CRP was measured as part of usual care, it was a standard measurement upon admission, but thereafter only in cases of clinical indication (i.e., if an infection was suspected or to monitor progress of a present infection). Consequently, too few participants had repeated CRP data with matching CPV data to be able to reliably examine their associations over time. This longitudinal association may be interesting to investigate in future research.

## 5. Conclusions

In conclusion, in contrast with our hypothesis, no statistically significant association was found between CRP and the capacity to perceived vitality ratio in hospitalized geriatric patients. Given the relatively selective and homogenous sample, verification of the findings in a larger, representative sample is required to explore whether CPV is a useful indicator of frailty or recovery capacity in hospitalized geriatric patients.

## Figures and Tables

**Figure 1 ijerph-20-06582-f001:**
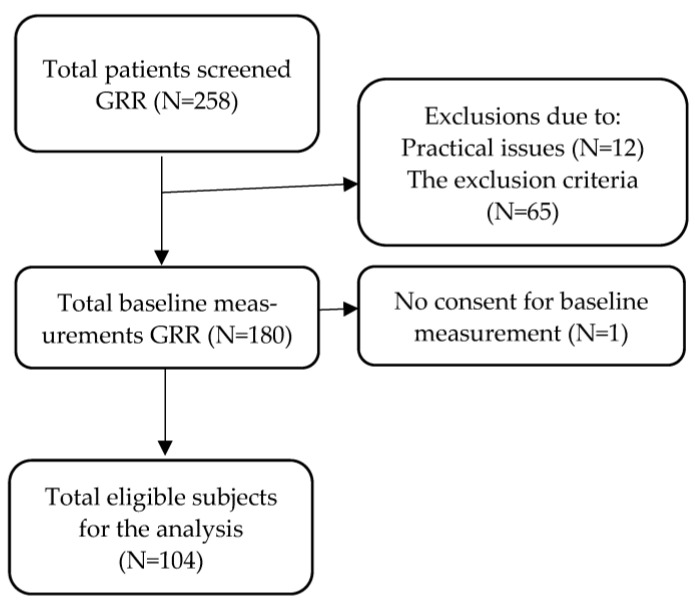
Flow chart: an overview of usable data for the GRR study.

**Table 1 ijerph-20-06582-t001:** Characteristics of the study sample.

Variables	Total Sample (*n* = 104)
Age (years, mean ± SD)	83.3 ± 7.5
Sex (% women)	55.8
Weight (kg)	72.0 ± 15.9
Clinical frailty scale (mean ± SD)	5.5 ± 1.4
Brief resilience scale (mean ± SD)	3.2 ± 0.8
No. of chronic conditions (median [IQR])	3 [2–5]
Length of stay (days, median [IQR])	7 [4–10]
Mortality during admission (% deceased)	26.0
Self-perceived fatigue (median [IQR])	6 [4–8]
Maximum grip strength (kPa, mean ± SD)	40.4 ± 17.4
Fatigue resistance (s, median [IQR])	24.5 [16–40]
Grip work (median [IQR])	675 [369–1193]
CPV ratio (median [IQR])	1.7 [0.9–4.2]
CRP (mg/L, median [IQR])	26.5 [4.8–60]
Time between CPV and CRP measurement (days, median [IQR])	1 [1, 2]
Reasons for admissions (n)	
Multiple admission reasons	15
General malaise	6
Musculoskeletal problems with a fracture	18
Musculoskeletal problems without a fracture	7
Cognitive problems	5
Cardio-vascular problems	5
Infections	28
Oncological reasons	1
Others	19

CPV, capacity to perceived vitality (higher ratio values indicate better recovery capacity); CRP, C-reactive protein; IQR, interquartile range; SD, standard deviation.

**Table 2 ijerph-20-06582-t002:** Cross-sectional association between CRP and CPV, GW and SPF, respectively.

	Sample Size	Regression Coefficient (95% Confidence Interval)
**Capacity to perceived vitality (CPV)**		
Unadjusted association	*n* = 86	−1.69 (−4.48 to 1.10)
Adjusted for confounders *	*n* = 86	−1.32 (−4.44 to 1.76)
Adjusted for confounders * and after imputation of missing data	*n* = 104	−0.33 (−4.00 to 3.34)
**Grip Work (GW)**		
Unadjusted association	*n* = 86	−187.50 (−679.59 to 304.54)
Adjusted for confounders *	*n* = 86	−116.71 (−676.73 to 433.30)
Adjusted for confounders * and after imputation of missing data	*n* = 104	25.53 (−478.23 to 529.30)
**Self-perceived fatigue (SPF)**		
Unadjusted association	*n* = 86	0.45 (−0.73 to 1.62)
Adjusted for confounders *	*n* = 86	0.58 (−0.76 to 1.91)
Adjusted for confounders * and after imputation of missing data	*n* = 104	0.57 (−0.64 to 1.77)

* Confounders included sex, the brief resilience scale, the clinical frailty scale and length of stay.

## Data Availability

Data are available upon request, as data collection is still ongoing.

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
