# Peer review of "Association between Muscle Fatigability, Self-Perceived Fatigue and C-Reactive Protein at Admission in Hospitalized Geriatric Patients"

_ijerph, 2023, doi:10.3390/ijerph20166582_

Round 1

Reviewer 1 Report

The estimation of individual biological reserves, the concepts of frailty, resilience, and intrinsic capacity are destined to be absolute protagonists in current and future research. Therefore, the topic you have chosen is of great relevance in geriatrics and gerontology. I would like to provide some considerations regarding your study:

I suggest modifying the title as "Association between muscle fatigability, self-perceived fatigue and C-reactive protein at admission in hospitalized geriatric patients" since the only inflammatory marker evaluated in the investigation is the CRP at admission. Therefore, it is advisable for the Authors to include the prefix "baseline" or "at hospital admission" in any speculation and explanation regarding CRP.

Furthermore, I suggest to better clarify the role of CRP in hospitalized patients. Of course CRP is an inflammatory biomarker, but it cannot be discerned in its clinical significance from infection, which is a common reason of hospitalization in older patients. Given that the study was carried out in hospitalized patients, CRP could reflect the infection state and disease severity rather than an inflammatory status. Therefore, association between CRP and inflammatory status is seems inconclusive. To properly verify this association, sepsis or any kind of infection should have been stated as exclusion criteria, and the study should have been powered accordingly. Otherwise, given that the study's main topic is resilience and intrinsic capacity, an association between the delta of CRP (highest CRP - lowest CRP during hospital stay) and CPV would have been more appropriate.

 Please comment and add as a limitation in the Discussion section.

Here other comments.

1.     The introduction focuses on the topic and the objective of the study. However, I suggest including a reference on resilience (preferably at line 36, right after "health stressors"). Similarly, it would be helpful to include references at line 41, after "perceived vitality ratio."

2.     Materials and methods are correctly described. The idea of introducing the use of a digitized grip work is original and very interesting. However, I have a few questions:

i)       Based on what objective criteria do you define a patient as having a life expectancy of less than two weeks?

ii)     In other studies on this topic (Knoop V. et al., Exp Gerontol, 2021; Bautmans, I. et al., J Gerontol A Biol Sci Med Sci, 2010), other tools such as MFI-20 were used to investigate the presence of subjective fatigue perception (SFP). It might be useful to specify the particular reason that led you to choose a single question with a response range of 0 (not tired at all) to 10 (extremely tired).

3.   The results are presented clearly. However, it could have been useful to include the discharge diagnosis of patients in the sample characteristics to understand the reasons for hospitalization, which could be related to both increased inflammatory markers and CPV. There is an extra closing parenthesis ")" in Table 1 under the "length of stay" row that needs to be corrected.

The novelty introduced by this study mainly concerns the idea of using a digitized grip work. However, with respect to the objective you have set, which is to study the association between CPV and inflammation, we cannot draw any definitive conclusions. It can be a valid starting point and a useful reference for anyone considering conducting a study investigating the correlation between CPV and inflammation. Therefore, I believe that, following the revision of the suggestions provided, it is worth publishing the results of this study.

Author Response

Dear reviewer, 

Kind regards, 

Ms. Carmen Hoekstra

Reviewer 2 Report

Various queries are raised with respect to the methodology as outlined below under relevant subheadings. Authors need to clarify these queries/comments.

Abstract

To be reconciled as per journal guidelines: add ‘Background’ instead of ‘Introduction’ in thr abstract.

An overview of the participant’s age/ gender with mean/SD or range values can be added to the methodology.

Keywords: Appropriate MeSH keywords may be used.

Introduction

The current state of the research field should be reviewed carefully and key publications cited.

The gap in the existing literature and need are to be brought out in the introduction in the preceding paragraphs.

More previous literature is need to be added.

Materials and Methods

Sample size calculation criteria should be given.

Please add the Institutional Review Board Statement and approval number.

Reference should be given for lines 113-116 and 121-124.

Give the name and version of the software used for statistical analysis.

A flow diagram of study participants enrolled in the study may be added for better comprehension. Mention the details of how the inclusion and exclusion criteria have been executed including the participants having missing values may be brought out in the flow diagram.

Results

The authors have to bring out how the results are commensurate with the aim of the study.

Discussion

Reference should be given. (Lines 202-203)

The authors have brought out how the study’s aim and findings will have practical value to the reader in the initial paragraphs.

Limitations of the study should be added.

Conclusion

This section needs to be added separately.

The language and grammar of the manuscript as a whole need little modification.

Author Response

(The authors gave the same response as above.)
